# A Quasiconformal-Based Geometric Model for Craniofacial Analysis and Its Application

**Ming-Hei Wong** [1,†], **Meixi Li** [1], **King-Man Tam** [1], **Hoi-Man Yuen** [2], **Chun-Ting Au** [2], **Kate Ching-Ching Chan** [2], **Albert Martin Li** [2] and **Lok-Ming Lui** [1,*]

1    Department of Mathematics, The Chinese University of Hong Kong, Hong Kong, China
2    Department of Paediatrics, Prince of Wales Hospital, The Chinese University of Hong Kong, Hong Kong, China
*    Correspondence: lmlui@math.cuhk.edu.hk
†    Current address: Department of Mathematics, The University of Tennessee, Knoxville, TN 37996, USA.

**Abstract:** We address the problem of craniofacial morphometric analysis using geometric models, which has important clinical applications for the diagnosis of syndromes associated with craniofacial dysmorphologies. In this work, a novel geometric model is proposed to analyze craniofacial structures based on local curvature information and Teichmüller mappings. A key feature of the proposed model is that its pipeline starts with few two-dimensional images of the human face captured at different angles, from which the three-dimensional craniofacial structure can be reconstructed. The 3D surface reconstruction from 2D images is based on a modified 3D morphable model (3DMM) framework. Geometric quantities around important feature landmarks according to different clinical applications can then be computed on each three-dimensional craniofacial structure. Together with the Teichmüller mapping, the landmark-based Teichmüller curvature distances (LTCDs) for every classes can be computed, which are further used for three-class classification. A composite score model is used and the parameter optimization is carried out to further improve the classification accuracy. Our proposed model is applied to study the craniofacial structures of children with and without the obstructive sleep apnoea (OSA). Sixty subjects, with accessible multi-angle photography and polysomnography (PSG) data, are divided into three classes based on the severity of OSA. Using our proposed model, our proposed model achieves a high 90% accuracy, which outperforms other existing models. This demonstrates the effectiveness of our proposed geometric model for craniofacial analysis.

**Keywords:** obstructive sleep apnoea; quasiconformal geometry; machine learning; preliminary disease diagnosis; image analysis; 3D facial model reconstruction

**MSC:** 30C75; 65K10; 65E10; 52C26; 65D19; 92C55

## 1. Introduction

Craniofacial morphometric analysis is a critical field in the study of human anatomy and biology. The utilization of mathematical models in craniofacial morphometric analysis has gained significant attention in recent years, providing a powerful tool for analysing and comprehending the intricate structural relationships of the human cranial and facial bones. These models not only aid in the diagnosis and treatment of syndromes associated with craniofacial anomalies, but also offer valuable insights into the genetic basis of craniofacial variability.

Conventional methods for craniofacial analysis are based on simple geometric measures on 2D X-ray images capturing the craniofacial bone structures [1–5]. In [1], the position of the hyoid bone and lateral parapharyngeal wall (LPW) thickness were used as anatomical markers to analyze a breathing disorder. In [4], the mentum–hyoid distance of

patients with a severe sleep disorder was longer compared to the other mild groups. Ref. [3] found a significant correlation between cephalometric data and apnoea–hypopnoea index (AHI) score severity in children with sleep disorder. The use of simple geometric measurements in conventional craniofacial analysis models has several drawbacks. One of the main limitations is that these measurements are based on 2D X-ray images, which can result in a loss of important information about the depth and thickness of cranial and facial bones. Additionally, 2D X-ray images may not accurately reflect the complex three-dimensional structure of the human head, leading to inaccurate measurements and predictions. Furthermore, the use of simple geometric measurements may not be suitable for capturing the geometric anomalies in more complex cases, such as those involving cranial and facial asymmetry or severe craniofacial anomalies. These limitations can result in incorrect morphometric analysis. Therefore, while conventional craniofacial structure analysis may be useful in some cases, it is important to use it in combination with other more advanced geometric models, to ensure accurate and comprehensive analysis of craniofacial structures.

Recently, many researchers also explored methods using mathematical imaging, machine learning, and artificial intelligence to improve the craniofacial analysis. Asghar's team [6] proposed a logistic classifier in craniofacial photographic features and a neural network in 2017 to automatically process frontal and profile photographs and classify patients as normal or diseased subjects. In 2018, Syed's team [7] created a deep learning model based on the VGGface data set [8] and 69 adult subjects for craniofacial analysis using a depth map of a person's facial scans. Curvatures have been proposed in recent years for craniofacial structure analysis. Kiaee's team [9] computes the L2 distance between two subjects using curvature.

Quasiconformal geometry has proved its effectiveness and accuracy in shape analysis for many years [10–12]. For example, ref. [10] developed an algorithm to automatically register hippocampal (HP) surfaces with complete geometric matching, avoiding the need to manually label landmark features. A recent study shows that based on quasiconformal geometry, X-ray images can be used to diagnose OSA [11]. In this work, we develop a new quasiconformal-based geometric model for craniofacial analysis. Our model is based on the local curvature information and Teichmüller mappings, which aims to provide a more comprehensive analysis of craniofacial structures. The proposed model starts with the acquisition of few two-dimensional images of the human face captured at different angles. These images are then used to reconstruct the three-dimensional craniofacial structure through a modified 3DMM framework. This reconstruction process is crucial as it allows us to compute geometric quantities around important feature landmarks. The landmarks can be manually labelled by experts according to different clinical applications or automatically delineated based on the local geometric information. The Teichmüller mapping is then applied to compute LTCDs for each class. These LTCDs serve as the geometric information for our three-class classification problem. To further improve the classification accuracy, a composite score model is used, and the parameters of the model are optimized.

In order to validate the effectiveness of our proposed model, we apply it to study the craniofacial structures of children with and without OSA. OSA is a common sleep-related breathing disorder characterized by recurrent episodes of partial or complete upper airway obstruction during sleep that leads to disruption of normal respiration and sleep patterns [13]. This disorder has a prevalence of 3% to 5% in children and is associated with cardiovascular, metabolic and neurocognitive sequelae [13]. Childhood OSA should be diagnosed promptly and treated early since some detrimental impacts, including adverse cardiovascular events and neurocognitive dysfunctions, can be induced by untreated OSA [13]. This calls for the development of an efficient and accurate diagnosis methodology. To test the effectiveness of our proposed mathematical model, we use a data set of 60 subjects, with accessible multi-angle photography and PSG data, and divide them into three classes based on the severity of OSA. Our proposed model achieves a high accuracy of 90%, outperforming other existing models. This demonstrates the efficacy of our proposed geometric model in the analysis of craniofacial structures.

The forthcoming section shall provide an exposition on the mathematical background of our proposed model.

## 2. Mathematical Background

Quasiconformal geometry and differential geometry are the key concepts that allow us to perform analysis on the subjects [14]. Quasiconformal geometry is used for image registration and differential geometry is used for shape analysis of 3D facial surfaces.

### 2.1. Quasiconformal Mapping

Quasiconformal mapping is a class of homeomorphisms on complex numbers that is orientation-preserving and has bounded conformality distortions. It is defined by the so called Beltrami equation.

**Definition 1.** *A mapping $f : \mathbb{C} \to \mathbb{C}$ is quasiconformal if there exists a complex-valued function $\mu_f(z)$ with $\|\mu_f(z)\|_\infty < 1$ such that*

$$\frac{\partial f}{\partial \overline{z}} = \mu_f(z)\frac{\partial f}{\partial z}. \tag{1}$$

An important special case is when $\mu_f(z)$ is equal to 0, then the equation above is reduced to the Cauchy–Riemann equation and so $f$ is conformal. The function $\mu_f$ is called the Beltrami Coefficient. It is a measurement of how far $f$ is from being conformal. Furthermore, note that the above equation gives us a way to calculate the Beltrami Coefficient by

$$\mu_f(z) = \frac{\partial f}{\partial \overline{z}} \Big/ \frac{\partial f}{\partial z}. \tag{2}$$

Clearly, given a mapping $f$, we can calculate the Beltrami Coefficient by computing partial derivatives. On the other hand, we can restore a mapping if we are given a Beltrami Coefficient. The mapping restored is unique if a stronger condition is imposed.

**Theorem 1** (Measurable Riemannian Mapping Theorem). *Let $\mu : \mathbb{C} \to \mathbb{C}$ be a Lebesgue measurable function with $\|\mu(z)\|_\infty < 1$ for all $z \in \mathbb{C}$, then there exists a quasiconformal homeomorphism $f$ mapping from unit disk to itself, which is in the Sobolev space $W^{1,2}(\mathbb{C})$ and satisfies the Beltrami equation in the distribution sense. Furthermore, if $\mu$ is stationary at 0, 1, and $\infty$, then the uniqueness of $f$ is guaranteed.*

By the last part of the theorem, normalization can be performed and the uniqueness result can then allow one to register between two images by homeomorphisms.

### 2.2. Teichmüller Map

Consider two Riemann surfaces $S_1, S_2$ in $\mathbb{R}^3$. A Beltrami differential $\mu(z)\frac{d\overline{z}}{dz}$ on a Riemann surface $S$ is an assignment to each chart $(U_\alpha, \phi_\alpha)$ on a $L_\infty$ complex-valued function $\mu_\alpha$, defined on the local parameter $z_\alpha$ such that $\mu_\alpha \frac{d\overline{z_\alpha}}{dz_\alpha} = \mu_\beta \frac{d\overline{z_\beta}}{dz_\beta}$ on the domain which is also covered by another chart $(U_\beta, \phi_\beta)$. An orientation-preserving diffeomorphism $f : S_1 \to S_2$ is called a quasiconformal mapping associated with the Beltrami Differential $\mu(z)\frac{d\overline{z}}{dz}$ if for any chart $(U_\alpha, \phi_\alpha)$ on $S_1$ and for any chart $(U_\beta, \psi_\beta)$ on $S_2$, the mapping defined by $f_{\alpha\beta} = \psi_\beta \circ f \circ \phi_\alpha^{-1}$ is a quasiconformal mapping.

In our application, all 3D facial surfaces reconstructed are simply-connected open surfaces, and hence they can be covered and represented by a single chart. Hence, the computation of quasiconformal mappings between any two facial surfaces can be performed through composition of mappings on the complex plane. A useful formula is given below to find the composition.

**Theorem 2.** *Let $f, g : \mathbb{C} \to \mathbb{C}$ be two quasiconformal mappings, then $g \circ f$ is also quasiconformal, and the Beltrami Coefficient of the mapping can be explicitly represented by*

$$\mu_{g \circ f}(z) = \frac{\mu_f(z) + \frac{\overline{f_z}}{f_z}\mu_g(f(z))}{1 + \frac{\overline{f_z}}{f_z}\overline{\mu_f(z)}\mu_g(f(z))}. \tag{3}$$

Back to our main problem. The tool used to do the image registration is called Teichmüller map (or T-map for short). The Definition of T-map is as follows:

**Definition 2** (Teichmüller Map)**.** *Let $f : S_1 \to S_2$ be a quasiconformal mapping with the Beltrami coefficient $\mu$. $f$ is a Teichmüller map (T-map) associated with the quadratic differential $q = \varphi dz^2$ where $\varphi : S_1 \to \mathbb{C}$ is a holomorphic function if its associated Beltrami Coefficient is of the form*

$$\mu(f) = k\frac{\overline{\varphi}}{|\varphi|} \tag{4}$$

*for some constant $k < 1$, and the quadratic differential $q$ must satisfy $q \neq 0$ and $\|q\|_1 = \int_{S_1} |\varphi| < \infty$.*

Next, we introduce another type of mapping that is closely related to T-map.

**Definition 3** (Extremal Quasiconformal Map)**.** *Let $f : S_1 \to S_2$ be a quasiconformal mapping. $f$ is said to be extremal quasiconformal if for any quasiconformal map $h : S_1 \to S_2$ isotopic to $f$ relative to the boundary, then*

$$K(f) \leq K(h) \tag{5}$$

*where $K(f)$ is the maximal quasiconformal dilation of $f$. Furthermore, it is uniquely extremal if the above inequality is strict when $h \neq f$.*

The following theorem will show the relation between the two mentioned classes of mapping.

**Theorem 3** (Landmark-matching Teichmüller map [15])**.** *Let $g : \partial\mathbb{D} \to \partial\mathbb{D}$ be an orientation-preserving diffeomorphism of $\partial\mathbb{D}$, where $\mathbb{D}$ is the unit disk. Furthermore suppose $g'(e^{i\theta}) \neq 0$ and $g''(e^{i\theta})$ is bounded. Let $\{l^k\}_{k=1}^n \in \mathbb{D}$ and $\{q^k\}_{k=1}^n \in \mathbb{D}$ be the corresponding interior landmark constraints. Then, there exists a unique Teichmüller mapping $f : (\mathbb{D}, \{l^k\}_{k=1}^n) \to (\mathbb{D}, \{q^k\}_{k=1}^n)$ matching the interior landmarks, which is the unique extremal extension of $g$ to $\mathbb{D}$. Here, $(\mathbb{D}, \{l^k\}_{k=1}^n)$ denotes the unit disk $\mathbb{D}$ with prescribed landmark points $\{l^k\}_{k=1}^n$.*

Hence, besides having uniform conformal distortion, T-maps are extremal in the sense that they minimize the maximal quasiconformal distortion, and so are useful to perform registration. Next, it can be seen that T-maps induce a natural metric, called the *Teichmüller distance* [16] that can be used to measure the difference between two shapes in terms of local geometric distortion.

**Definition 4** (Teichmüller distance)**.** *For every $i$, let $S_i$ be the $i$th Riemann surface with landmarks $\{p_i^k\}_{k=1}^n$. The Teichmüller distance between $(f_i, S_i)$ and $(f_j, S_j)$ is given by*

$$d_T((f_i, S_i), (f_j, S_j)) = \inf_{\varphi} \frac{1}{2}\log K(\varphi) \tag{6}$$

*where $\varphi : S_i \to S_j$ varies over all quasiconformal mappings with $\{p_i^k\}_{k=1}^n$ corresponding to $\{p_j^k\}_{k=1}^n$, which is homotopic to $f_j^{-1} \circ f_i$, and $K$ is the maximal quasiconformal dilation.*

*2.3. Curvature*

After performing registration using T-map, the Gaussian and Mean curvature of specific points are extracted and statistical testing will be conducted on the curvature data.

2.3.1. Shape Operator

Let $M$ be a regular orientable surface on $\mathbb{R}^3$ with respect to the Normal $\mathbf{N}$. The *shape operator* can be defined as follows.

**Definition 5** (Shape Operator)**.** *Let $p \in M$ be a point and $v \in T_p(M)$, where $T_p(M)$ is the tangent space of $M$ at $p$. Let $\alpha(t) : (-\epsilon, \epsilon) \to M$ be a curve on $M$ such that $\alpha(0) = p$ and $\alpha'(0) = v$. Then, the shape operator $S_p(v)$ is defined as*

$$S_p(v) = -\frac{d}{dt}(\mathbf{N}(\alpha(t)))|_{t=0}. \tag{7}$$

Now, consider a parametrization $X(u, v)$ of the surface $M$, the tangent space at any point can be spanned by the set of vectors $\beta = \{\mathbf{X}_u, \mathbf{X}_v\}$ since $M$ is regular. Now, also note that $S_p$ map vectors from $T_p(M)$ to itself. We then have the matrix representation of the shape operator.

**Theorem 4.** *Suppose $S_p(\mathbf{X}_u) = a_1^1 \mathbf{X}_u + a_1^2 \mathbf{X}_v$ and $S_p(\mathbf{X}_v) = a_2^1 \mathbf{X}_u + a_2^2 \mathbf{X}_v$. The matrix representation of the shape operator with respect to the basis $\beta$ is given by*

$$[S_p]_\beta = \begin{pmatrix} a_1^1 & a_2^1 \\ a_1^2 & a_2^2 \end{pmatrix}. \tag{8}$$

2.3.2. Gaussian and Mean Curvature

Using the notion of shape operator, curvatures can be defined as follows.

**Definition 6** (Gaussian and Mean Curvature)**.** *The Gaussian curvature $K(p)$ of the surface $M$ at $p$ is the determinant of $S_p$. The Mean curvature $H(p)$ of the surface $M$ at $p$ is equal to $\frac{1}{2} Tr(S_p)$.*

Note that this matrix $S_p$ must be diagonalizable. If it is written in the diagonalized form $S_p = P^{-1} \begin{pmatrix} k_1 & 0 \\ 0 & k_2 \end{pmatrix} P$, it can be obtained that $K = k_1 k_2$ and $H = \frac{k_1 + k_2}{2}$. The numbers $k_1$ and $k_2$ are called the *principle curvatures* at that point.

Note that Gaussian Curvature is an intrinsic measure while Mean curvature is an extrinsic measure. Both curvatures are incorporated in the model for more comprehensive testing.

## 3. Proposed Model

This section discusses about the proposed model for preliminary classifying OSA patients. The first subsection talks about the image registration method based on a 3D morphable model and Teichmüller map. Then, geometric distortions of the specific landmarks are calculated based on quasiconformal geometry to generate a feature vector for each subject, which is discussed in the second subsection. With the discriminating feature vectors, an OSA classification model is proposed in the last subsection.

*3.1. The 3D Surface Reconstruction from 2D Images*

The 3D surface reconstruction from 2D images captured from multiple angles is a crucial step in accurate shape analysis. This process provides a more comprehensive representation of the object's shape, including its depth and three-dimensional structure, which is essential for analysing and understanding its geometric patterns. The use of multiple

images captured from different angles helps to overcome the limitations of 2D imaging, such as occlusions, and provides a more complete representation of the surface. Furthermore, 3D surface reconstruction enables the use of advanced shape analysis techniques, such as surface registration and shape comparison. As such, the first step of our proposed model is to convert 2D images capturing the human face from multiple angles into one 3D facial surface. This procedure is carried out using a deep neural network that regresses the coefficients of the 3DMM face model.

The 3D reconstruction model is based on the 3D morphable face model, commonly referred to as a 3DMM. The 3DMM is a statistical model of the shape and appearance of human faces. The model is created by analysing a large number of 3D scans of human faces, which are then used to generate a set of parameters that can be used to generate an infinite number of 3D faces. With the 3DMM, the face shape $\mathbf{S}$ and the texture $\mathbf{T}$ can be represented as:

$$\mathbf{S} = \mathbf{S}(\boldsymbol{\alpha}, \boldsymbol{\beta}) = \bar{\mathbf{S}} + \mathbf{B}_{id}\boldsymbol{\alpha} + \mathbf{B}_{exp}\boldsymbol{\beta}$$
$$\mathbf{T} = \mathbf{T}(\boldsymbol{\delta}) = \bar{\mathbf{T}} + \mathbf{B}_t\boldsymbol{\delta} \tag{9}$$

where $\bar{\mathbf{S}}$ and $\bar{\mathbf{T}}$ are the average face shape and texture; $\mathbf{B}_{id}$, $\mathbf{B}_{exp}$, and $\mathbf{B}_t$ are the PCA bases of identity, expression, and texture, respectively; $\boldsymbol{\alpha}$, $\boldsymbol{\beta}$, and $\boldsymbol{\delta}$ are the corresponding coefficient vectors for generating a 3D face. The Basel Face Model [17] is used for $\bar{\mathbf{S}}$, $\mathbf{B}_{id}$, $\bar{\mathbf{T}}$, and $\mathbf{B}_t$, and the expression bases $\mathbf{B}_{exp}$ in [18] is used. In order to train the backbone deep neural network for obtaining the 3D face from a 2D image, every 3D face in the training data set is matched with a corresponding 2D image. These 2D images are taken from different angles, including 0 degree (frontal view), 30 degrees, 45 degrees, −30 degrees, and −45 degrees. A subset of the bases is selected based on the five images, resulting in $\boldsymbol{\alpha} \in \mathbb{R}^{80}$, $\boldsymbol{\beta} \in \mathbb{R}^{64}$ and $\boldsymbol{\delta} \in \mathbb{R}^{80}$. Given the training data with pairs of 2D images and their associated 3D shapes, a deep neural network regresses the coefficient vector $\mathbf{x} = (\boldsymbol{\alpha}, \boldsymbol{\beta}, \boldsymbol{\delta}, \boldsymbol{\gamma}, \mathbf{p}) \in \mathbb{R}^{239}$. In this work, we adopt the method in [19] to obtain the coefficient vector $\mathbf{x}_j$ associated with an image $I_j$ capturing the human face from one angle.

Now, to integrate all images from multiple angles to obtain a more accurate 3D model, the following procedure is carried out. For each image $I_j$, the spatial dependent weight $\mathbf{w}_j$ is assigned. The weight depends on how informative the image $I_j$ for the 3D reconstruction of the 3D model. For images of poor image quality, we assign a smaller weight. The weight of each image is determined by its sharpness and information content. Specifically, the weight for each image is represented as a vector with the same size as the number of vertices on the 3D reconstructed surface. Therefore, each vertex on the 3D surface is assigned a weight associated with each image. The image sharpness $\mathcal{S}(I)$ of an image $I$ is calculated using the Laplacian operator, which is defined as $\mathcal{S}(I) = ||\Delta I||_1$, where $\Delta$ represents the Laplacian kernel. By convolving $I$ with the Laplacian kernel, $\Delta I$ captures the features and edges of objects in the image. The magnitude of $\Delta I$ is higher for sharper images, resulting in a larger value of $\mathcal{S}(I)$ for sharper images. In addition to image sharpness, the image information content is also taken into account. The image information refers to the reliability of the 2D image in providing information for determining the 3D coordinates of a point on the reconstructed face. For example, a frontal image (captured at 0 degree) provides reliable information for the central region of the 3D face. Therefore, the image information weights associated with vertices of the central region are defined as 1. However, if an image cannot capture a region of the 3D face, the image information weights associated with vertices of the occluded region are defined as 0. The final weight for each vertex associated with each image is calculated as the product of its image sharpness and image information weight. For spatial position with a poor quality, such as the existence of occlusions, a smaller weight associated with that particular position can be assigned. The final 3D reconstructed model is given by:

$$\mathbf{S} = \sum_{j=1}^{N} \mathbf{w}_j \odot \mathbf{S}(\boldsymbol{\alpha}_j, \boldsymbol{\beta}_j) = \sum_{j=1}^{N} \mathbf{w}_j \odot (\bar{\mathbf{S}} + \mathbf{B}_{id}\boldsymbol{\alpha}_j + \mathbf{B}_{exp}\boldsymbol{\beta}_j), \tag{10}$$

where $\odot$ refers to the pointwise multiplication of two vectors and $N$ is the number of images from multiple angles.

Figure 1 gives an illustration of the 3D reconstruction from 2D images.

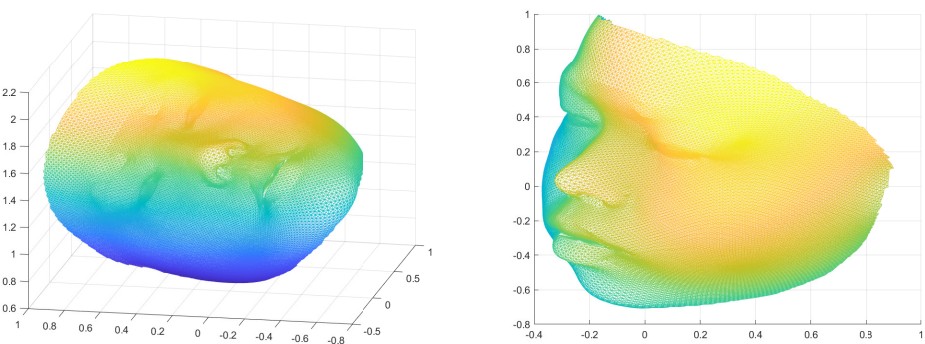

**Figure 1.** Demonstration of the 3D facial model reconstructed from 2D photos.

### 3.2. Landmark-Based Teichmüller Curvature Distance (LTCD)

In order to train a three-class classification machine, the distances of each data from each classes have to be defined. The choice of distances play an important role for the classification accuracy. Computing distances between data and classes is important for classification because it provides a way to measure the similarity between each datapoint and each class. This information is crucial for assigning each datapoint to the most appropriate class, and can be used to determine the class membership of each datapoint based on the closest match. Furthermore, the computation of distances between data and classes can also provide valuable insights into the underlying structure and relationships between the data and the classes, which can be useful for developing better classification models and improving accuracy.

In this work, our choice of distances is based on surface geometry on selected feature landmarks, which will be described in this subsection.

#### 3.2.1. Surface Registration

First of all, LTCD is defined as the distance of each data to each class. In order to define distances between different data, a one-to-one correspondence between data must be obtained. In this work, a landmark-matching registration model by computing the Teichmüller mapping that matches facial landmark features is adopted to compute the mutual correspondence between subjects [14]. The Teichmüller mapping can be formulated as an extremal mapping minimizing the local geometric distortion. As such, the problem of computing the Teichmüller mapping can be converted into an optimization problem

$$(v, f) = \underset{v:D_1 \to \mathbb{C}}{argmin} \{ \|v\|_\infty + \|\nabla v\|_2 \} \tag{11}$$

subject to: (i) $v = \mu(f)$ and $\|v\|_\infty < 1$; (ii) $v = k\dfrac{\overline{\varphi}}{|\varphi|}$ for some constant $k$ and holomorphic function $\varphi : D_1 \to \mathbb{C}$; and (iii) $f$ satisfies certain boundary condition and/or landmark constraints. The optimization problem can be solved by the Quasiconformal Iteration (QC) using the Linear Beltrami Solver (LBS).

The best quasiconformal mapping associated with a given Beltrami coefficient can be obtained with the use of LBS. An algorithm (QC iteration for open surfaces) can be used to obtain the extremal mapping $f$ [14]. The main idea of the algorithm is to iteratively search for the optimal Beltrami Coefficient associated with $f$. Using the optimal Beltrami coefficient, the desired extremal mapping $f$ can be easily reconstructed using the LBS. One benefit of the application of the quasiconformal registration is that the effect of global scaling, global rotation, and global translation is minimized.

Two 3D triangular meshes are used as input as well as the expected boundary condition. An optimal Beltrami coefficient $v$ and the Teichmüller mapping $f$ are output as a result. The algorithm follows a three-steps approach.

Firstly, the initial mapping is set

$$f_0 = LBS_{LM}(\mu_0 := 0) \tag{12}$$

and fix the initial Beltrami coefficient $v_0 = \mu(f_0)$.

Secondly, we can iteratively compute

$$\begin{aligned} \mu_{n+1} &:= \mathcal{A}(\mathcal{L}(v_n)) \\ f_{n+1} &:= LBS_{LM}(\mu_{n+1}) \\ v_{n+1} &:= \mu(f_{n+1}) \end{aligned} \tag{13}$$

where $\mathcal{A}$ is the averaging operator, $\mathcal{L}$ is the Laplacian operator, and $LBS$ is the Linear Beltrami Solver. Lastly, this alternating process continues unless $||v_{n+1} - v_n|| < \epsilon$. For more details about the algorithm of the registration model, readers are referred to [14].

Once the pairwise surface registration is computed, every craniofacial surfaces can be represented by a triangulation mesh with the same connectivity and the same number of vertices with correspondence.

### 3.2.2. Surface Geometric Feature Vector

Our next step is to obtain a geometric feature vector for each surface, which can be used to define distances between surfaces. The feature vector should comprise the most crucial geometric information that accurately characterizes the surface, as determined by the specific application and its relevant discriminating power. We construct the geometric feature vector for our classification machine as follows. Suppose there are $N$ subjects, in which the first $N/3$ subjects are in the control class (class 0), the second $N/3$ subjects are in the mild OSA class (class 1), and the last $N/3$ subjects are in the moderate-to-severe OSA class (class 2). In this work, a method to construct a feature vector containing Gaussian and Mean curvature data is proposed.

Each of the 3D images is registered to 35,710 indices by the above registration model. In this work, we propose to consider geometric information around some feature landmarks only. This avoids misleading geometric information from unimportant region to interrupt the classification result. Therefore, $n$ landmarks around are selected either manually or automatically according to each practical applications (see Figure 2 for an illustration).

For each landmark point that is selected manually, the closest $w^2$ points based on Euclidean distances are chosen as feature points as well, where $w$ is adjusted in the iterations of the algorithm. Therefore, $28w^2$ points are considered for each subject. Some of the points may be duplicate in this process and duplicate entries are removed. After reordering the index in ascending order, a collection of these landmark points can be constructed and denoted as $S$, where $S_j$ represents the $j$-th item in the collection $S$. Figure 3 provides a visual representation of the outcome obtained through automatic landmark selection.

To obtain corresponding curvature data, a standard algorithm is used. The normalized Gaussian and Mean curvature are denoted as $K$ and $H$, respectively, which are combined to form a feature vector $C_i$ for the $i$-th subject, that is, $C_i = [K_i, H_i]$. More specifically, we normalize both the mean curvature and Gaussian curvatures to be within the interval $[0, 1]$. Denote the length of the feature vector by $L$ and $C_i(j)$ be the $j$-th entry of the feature vector.

To augment the discriminating power of the feature vector, a $t$-test incorporating the bagging predictors is used to select a certain percentage of features with the highest discriminating power [11,20]. In the general $t$-test, a probability $p_j$ called the $p$-value is computed for each feature point $C_i(j)$ which evaluates the power of the feature in discriminating the given three classes. The bagging predictors strategy uses a leave-one-out scheme to improve the stability of the $t$-test. Specifically, $N$ tests are performed, and each test is performed on all the subjects excluding the $i$-th one. This gives the $p$-value $p_j^i$ for

feature point $j$ in the $i$-th iteration. After all the tests, the $p$-value of each feature is calculated by $p_j = \min_i \ p_j^i$. According to the $t$-test, the discriminating power of each feature increases as its $p$-value decreases. So, $\tilde{L}$ of the features with the highest discriminating power are $\tilde{L}$ of the features with the smallest $p$-values, where $\tilde{L}$ is adjusted in the iterations of the algorithm. Based on the $p$-values, the features with low discriminating power can be removed from our classification machine by remaining only the $\tilde{L}$ features with the highest discriminating power given a certain percentage. Figure 4 provides a visual depiction of the bagging outcome.

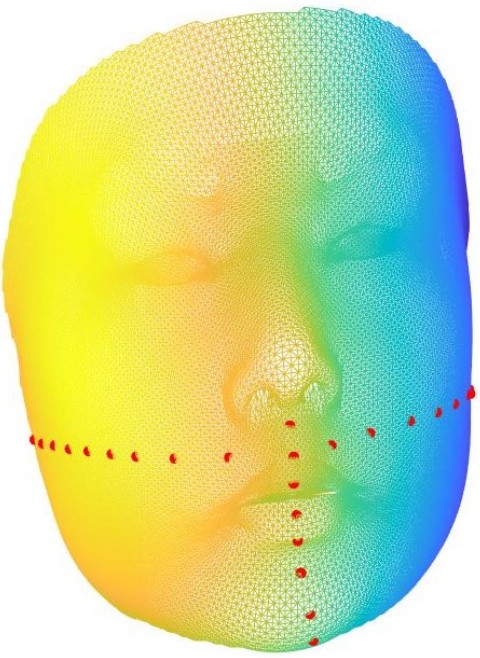

**Figure 2.** Demonstration the facial landmark points (red dots) superimposed on a sample 3D face model.

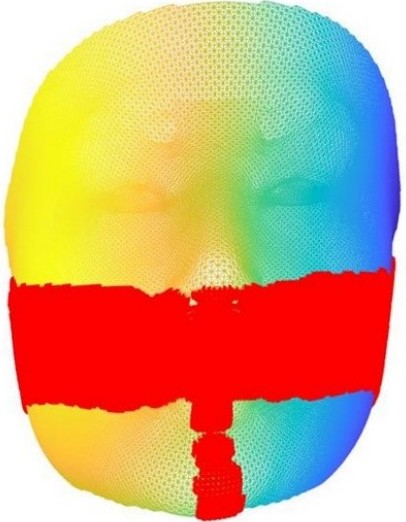

**Figure 3.** Automatic landmark selection by the Euclidean Distance Method.

In this work, our model uses the discriminating feature vector $\tilde{C}_i$ for each craniofacial structure.

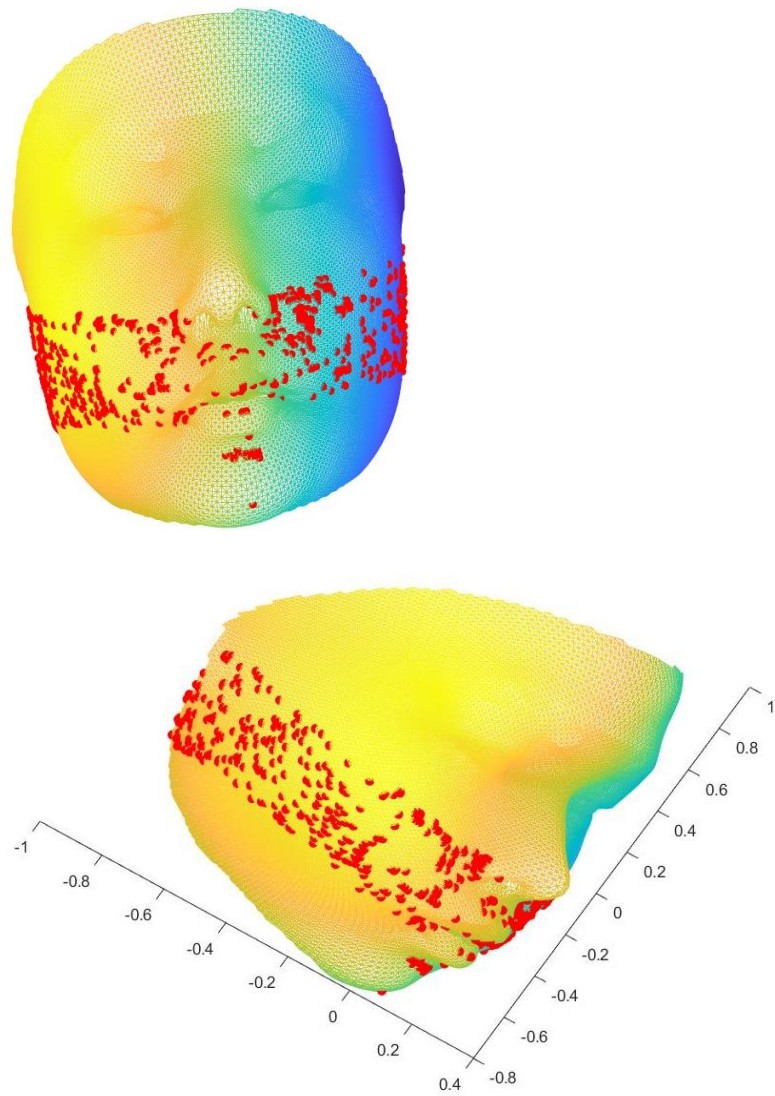

**Figure 4.** Landmarks having high classification power chosen by the bagging algorithm.

### 3.2.3. LTCD Computation

Based on the above preparation, the LTCD can be constructed. In this work, a simple $L^2$-norm is applied for three-class classification. The idea is that subjects from each class possess an analogous geometrical structure to the skull. The subjects can then be classified by the difference between the template feature vector and subjects. Note that each surface meshes to be analysed are in pairwise correspondence with each others. This ensures that each entry of the feature vector for a given subject can be compared directly to the corresponding entry for another subject, without the need to adjust for point mismatching issues. Evidently, the choice of surface correspondence is critical to the success of this analysis, as it determines the quality of the comparison between subjects. In our model, we use the Teichmüller mapping, which minimizes the conformality distortion and hence, the local geometric distortion. This ensures that the comparison between subjects is accurate and meaningful, with the most important geometric information being captured and considered in the analysis. Since the discriminating feature vector is obtained by bagging, the median of the feature vectors among each class can be generated. The median of each class is used instead of the mean to avoid extreme data. If a larger set of data is adopted, the mean may replace the median to achieve a higher accuracy.

Let $V_i = \{\tilde{C}_k \mid \text{Subject } k \text{ is in class } i\}$, for $i = 0, 1, 2$, be the three collections of the trimmed feature vectors belongs to the three classes. Then, the template feature vectors $T_0$, $T_1$, $T_2$ of each class can be defined as

$$T_i(j) = \underset{v \in V_i}{\text{median}}\, v(j) \tag{14}$$

for $i = 0, 1, 2$ and $j = 1, 2, ..., \tilde{L}$. Since there are 3 classes, the distances to $T_0$, $T_1$ and $T_2$ are defined for each subject in the $L^2$ sense as follows.

$$d_{ij} = \|\tilde{C}_i - \tilde{T}_j\| \tag{15}$$

$d_{ij}$ is called the LTCD of subject $i$ to the class $j$.

If the subject $i$ has a smaller distance to the template vector of class $j$, it is more likely that the subject belongs to class $j$. We remark that the choice of landmark points plays a crucial role in the analysis of surface meshes and can greatly affect the definition of geometric features. This in turn affects the landmark-based Teichmüller curvature distance, which is used to quantify the similarity between meshes. The choice of landmarks can be made either manually or automatically, depending on the specific practical application. It is important to carefully consider the choice of landmarks, as it can greatly impact the accuracy and validity of the analysis.

Last but not least, since we have three classes, it is difficult to judge when the magnitude of two of the distance vectors are close. A non-linear balancing term is added to increase the gap between classes, and the details will be discussed in the later section.

### 3.3. Composite Score Model and Parameter Optimization

In our work, since the length of the feature vector is rather long (usually with at least 500 coordinates), it is not suitable to adapt the commonly used classification algorithm, such as supporting vector machine (SVM) or $K$-nearest neighbour (KNN) algorithm. These algorithms are sensitive to a small perturbation of the feature vector, and also to outliers. Since medical data on class-1 patient is insufficient, training with outliers is not desirable. In particular, according to the opinion of the medical doctors, it is very difficult for them to distinguish class-0 and class-1 subject. In view of this situation, we need a model that can tolerate a margin of error whilst maintaining overall accuracy.

We propose the following composite score model, with the set of parameter $\{\alpha, \beta, \delta, \gamma\}$. These composite scores are based on the $l_2$ distance between the subject feature vector and the template feature vectors which can be computed fast and easily across machines.

**Definition 7** (Composite Score Model). *For the i-th subject, the set of composite score $\{S_{i0}, S_{i1}, S_{i2}\}$ is defined by*

$$\begin{aligned}
S_{i0} &= \alpha(d_{i0})^\delta - \beta(d_{i2})^\gamma \\
S_{i1} &= \alpha(d_{i1})^\delta - \left(\frac{\beta}{2}\right)(d_{i0} + d_{i2})^\gamma. \\
S_{i2} &= \alpha(d_{i2})^\delta - \beta(d_{i0})^\gamma
\end{aligned} \tag{16}$$

The same $\alpha, \beta, \delta, \gamma$ for the three composition scores instead of $\alpha_i, \beta_i, \delta_i, \gamma_i$ for each class since the sample size is small and this can avoid over-fitting. Furthermore, this composition score based on $l_2$-norm is considering all the coordinates of the feature vector as a whole, and so the influence from a particular extreme coordinate of a feature vector is minimized.

Furthermore, the criterion for classification using the composite scores is defined as follows.

**Definition 8** (Criterion for Classification). *Suppose the set $\{S_{i0}, S_{i1}, S_{i2}\}$ contains distinct numbers. The i-th subject is classified into the class p if*

$$S_{ip} = \min_{k} S_{ik}. \tag{17}$$

For the intuition behind the model, it is based on the fact if the *i*-th subject is closer to the template surface of class *k*, the value of $d_{ik}$ will be the smallest among $d_{i0}, d_{i1}, d_{i2}$. Furthermore, if the *i*-th subject is of class *k*, the distance between the subject and the other two classes will be bigger. By considering this two push-and-pull factors, the above composite score model is defined.

We now use the following illustration to help understand the composite score model.

Graphical Illustration of the Composite Score Model

We can interpret the class "0, 1, 2" as the degree of severity of the disease. If we plot the template feature vector $T_0, T_1, T_2$ in $\mathbb{R}^{\tilde{L}}$, we should expect that $T_1$ should be lying on somewhere near the midpoint of $T_0$ and $T_2$, as shown in Figure 5.

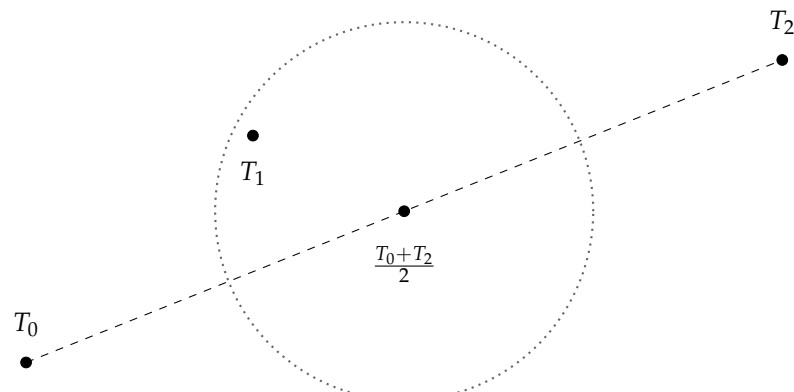

**Figure 5.** Reasonable expectation of the distribution of the template feature vector $T_0, T_1, T_2$.

Let us say we have a new subject *K* and we want to classify it, as shown in Figure 6. Suppose $d_{k0} = d_{k1}$. That means the new subject is equidistant from the template feature vectors $T_0$ and $T_1$.

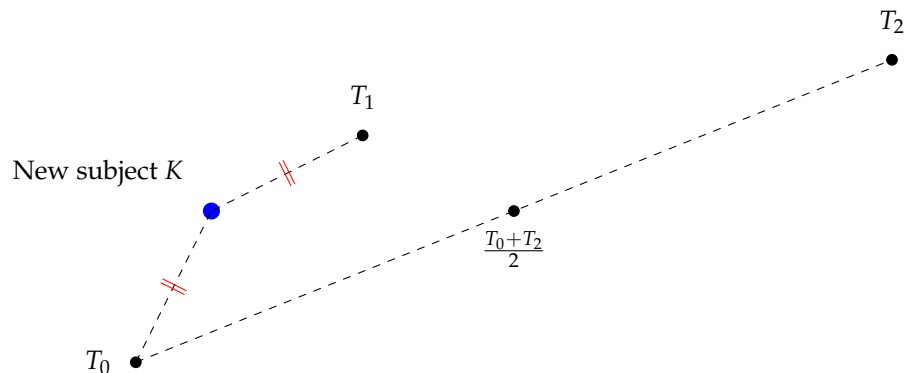

**Figure 6.** A new subject *K* that is equidistant from $T_0$ and $T_1$ is needed to be classified.

In this case, it is difficult to classify *K*. However, we can consider $d_{k2}$ as a "push factor", as shown in Figure 7. Note that $d_{k2}$ is large.

For example, we can define $s_{k0} = d_{k0} - d_{k2}$, $s_{k1} = d_{k1} - \frac{d_{k0}+d_{k2}}{2}$ as two "composition scores". This is a special case of our original composition score model Definition 7 with $(\alpha, \beta, \delta, \gamma) = (1, 1, 1, 1)$. Now, since $s_{k0} < s_{k1}$, we can classify the new subject as class 0.

Going back to our original composition score model, to simplify and speed up the process, we fix $\alpha = 1$ and iterate $\beta$ in $(0, 2)$ with 0.2 difference each time. The parameters $\alpha$, $\beta$ are used to balance the two push-and-pull factors. Furthermore, to provide a more flexible variation space, $\delta$ and $\gamma$ are added for non-linear approximation/optimization. It can provide a different approach for improving accuracy. To avoid over-fitting, $\gamma$ and $\delta$ are only chosen in $(0.6, 1.4)$ with 0.1 difference each time.

After iteration, we obtain the best result with

$$(\alpha, \beta, \delta, \gamma) = (1.0, 0.4, 1.2, 1.3)$$

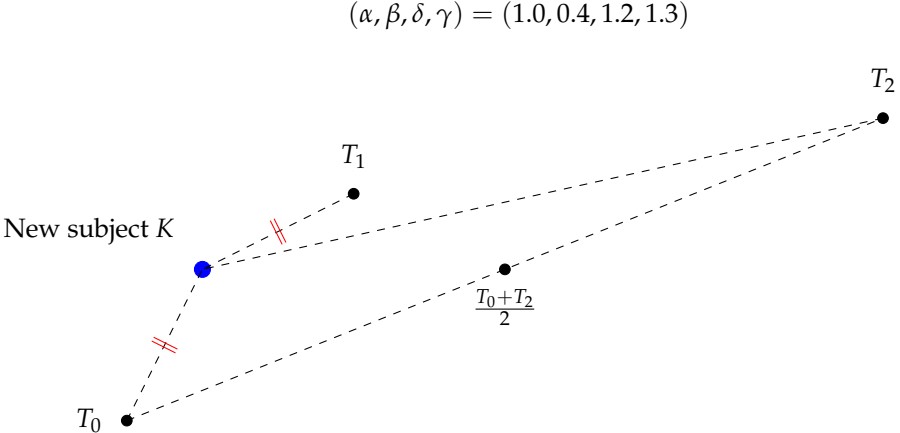

**Figure 7.** Take $d_{k2}$ into the consideration of classification.

## 4. Experiments Results

In order to validate the effectiveness of our proposed model, we apply it to study the 60 craniofacial structures of children with and without OSA. The subjects are grouped into three classes according to the severity of OSA. Class 0 refers to the control group, Class 1 refers to the mild group, and Class 2 refers to the severe group. We report the classification result in this section, as well as comparing our results with other methods.

### 4.1. Data Information

The present section is dedicated to presenting the data pertaining to the subjects, along with an exposition on polysomnography and multi-angle photography.

#### 4.1.1. Subjects

This work was based on 60 normal and OSA subjects of Chinese children recruited for sleep studies in the Prince of Wales Hospital, with accessible multi-angle photography and PSG. The same classification criterion for OSA is also used in [11]. Obstructive apnoea–hypopnoea index (OAHI) is the total number of obstructive and mixed apnoeas and hypopnoeas per hour of sleep. The group of OSA patients can be classified into three types, non-OSA, mild OSA, and moderate-to-severe OSA, which are defined by OAHI $\leq 1/h$, OAHI between $1/h$ and $5/h$, and OAHI $\geq 5/h$, respectively, where $h$ represents hour. Photos of children's faces from different angles were taken on the same day of admission. Patients with surgical treatment for OSA before photography and PSG, congenital or acquired neuromuscular disease, genetic or syndromal disease, craniofacial abnormalities, or obesity secondary to an underlying cause were excluded. In order to obtain the local deformation information, 3D facial models were reconstructed based on 2D photos of those subjects using a 3DMM machine learning model. To study OSA, 28 landmarks on the lower-half face were selected for classification.

#### 4.1.2. Polysomnography

The nocturnal PSG was conducted at the Prince of Wales Hospital. A model SiestaTM ProFusion III PSG monitor (Compumedics Telemed, Abbotsford, Victoria, Australia) was adopted to record the following parameters: electroencephalogram (F4/A1, C4/A1,

O2/A1), bilateral electro-oculogram, electromyogram of mentalis activity and bilateral anterior tibialis. Inductance plethysmography was used to measure respiratory movements of the chest and abdomen. Electrocardiogram and heart rate were constantly recorded from two anterior chest leads. An oximeter was used to estimate arterial oxyhaemoglobin saturation (SaO2) with a finger probe. A nasal catheter connected to a pressure transducer was placed at the anterior nares to measure respiratory airflow pressure signal. The absence of airflow was also detected by an oronasal thermal sensor. Snoring was recorded by a snoring microphone placed near the throat. A body position sensor was used to monitor body position.

An adequate nocturnal PSG requires at least 6 hours of total recorded sleep time. Respiratory events, including obstructive apnoeas, mixed apnoeas, central apnoeas, and hypopnoeas, were evaluated based on the recommendation from the American Academy of Sleep Medicine (AASM) Manual for the Scoring of Sleep and Associated Events Respiratory effort-related arousals (RERAs) were scored when the amplitude of nasal pressure signal fell under a half of baseline with flattening of the nasal pressure waveform, accompanied by snoring, noisy breathing, or signals of increased effort of breathing. A respiratory event was scored when it lasted $\geq 2$ breaths regardless of its duration. Arousal is defined as a sharp shift in electroencephalographic (EEG) frequency during sleep, including theta, alpha, and/or frequencies larger than 16 Hz but not spindles, lasting for 3 to 15 s. In rapid eye movement (REM) sleep, arousal is scored only when submental electromyogram (EMG) amplitude increases concurrently.

A senior research assistant who has Registered Polysomnographic Technologist (RPSGT) qualification and experience in performing paediatric PSG was responsible for the PSG scoring and reporting. The research assistant had no access to other assessment data of the subjects.

### 4.1.3. Multi-Angle Photography

Photos of children's faces from multiple angles were taken on the same day of admission to overnight PSG. To reconstruct the 3D facial mesh, three 2D images are used for each subject. These 2D images are taken from different angles, including 0 degree (frontal view), 45 degrees, and $-45$ degrees. An iPhone was used this time.

### *4.2. Results of the Main Model*

This work is based on 60 subjects consisting of 20 control subjects, 40 OSA subjects, including 20 mild OSA subjects and 20 moderate-to-severe OSA subjects. The accuracy of the prediction is defined by the total number of subjects correctly classified as non-OSA subjects and OSA subjects divided by the number of all subjects involved, regardless of severity. All subjects are used for training the parameters.

### 4.2.1. Binary-Class Model

For simplicity, a binary model is built with only control subjects and moderate-to-severe OSA subjects. As mentioned in Section 3.2, the $p$-value of each feature is calculated. According to the $t$-test, feature having higher discriminating power will have smaller $p$-value. So, $\tilde{L}$ of the features with the highest discriminating power are $\tilde{L}$ of the features with the smallest $p$-values. When using 0.51% of the features with the highest discriminating power, the accuracy is the highest. With 592 discriminating features, the model achieves 97.5% accuracy, with 100% sensitivity and 95% specificity. The optimal cut of the distances is 0.011824. Figure 8 shows the relation of the number of feature used and the classification accuracy.

### 4.2.2. Three-Class Model

To develop a model suitable for all subjects, mild subjects are also included in the revised version. When using 2.4% of the features with the highest discriminating power and $(\alpha, \beta, \delta, \gamma) = (1.0, 0.4, 1.2, 1.3)$, respectively, the accuracy is the highest. With 567

discriminating features, the model achieves 90.0% accuracy, with 100% sensitivity and 70% specificity. Table 1 shows the corresponding parameter values and the training accuracy when the number of feature is fixed to be 500, 1000, and 1500.

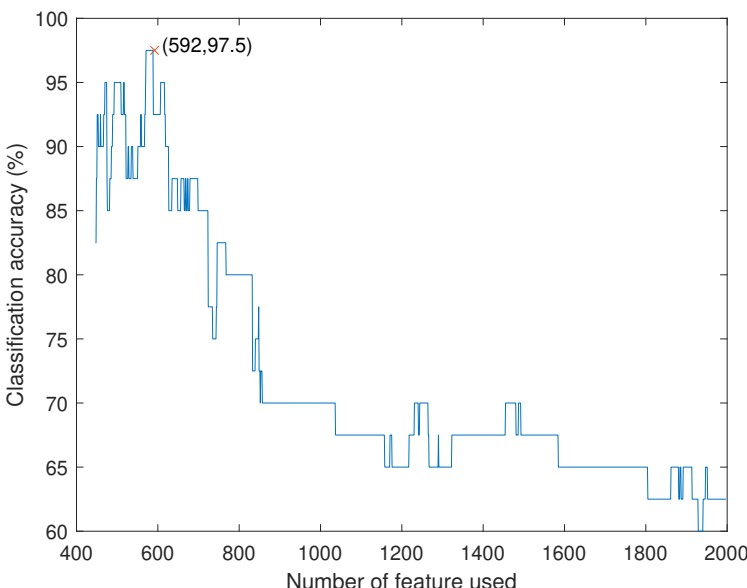

**Figure 8.** Classification accuracy against number of feature used for the binary-class model.

**Table 1.** Statistics of the classification accuracy of the proposed model including all three classes using different numbers of features.

| Number of Feature | $(\alpha, \beta, \gamma, \delta)$ | Sensitivity | Specificity | Accuracy |
|---|---|---|---|---|
| 500 | (1.0, 0.8, 1.2, 1.1) | 90.00% | 80.00% | 86.67% |
| 1000 | (1.0, 0.4, 0.6, 0.8) | 67.50% | 80.00% | 71.67% |
| 1500 | (1.0, 0.2, 0.6, 0.8) | 80.00% | 85.00% | 81.67% |

Figure 9 shows the relation between the number of feature used and the training accuracy of the three-class model.

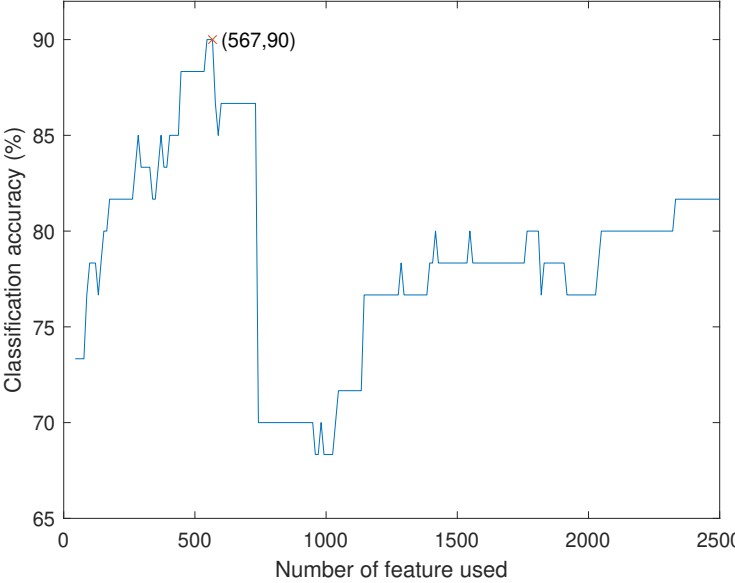

**Figure 9.** Classification accuracy against number of feature used for the three-class model.

### 4.2.3. Models with Unsatisfactory Results

Methods using geodesic distances of 3D facial models reconstructed from 2D images and distances between selected landmarks of 2D images are also tried among non-OSA and moderate-to-severe OSA patients. The method based on distances between selected landmarks of 2D images achieves only 57.5% accuracy, with 80% sensitivity and 35% specificity. The method based on geodesic distances of 3D faces reconstructed from 2D images achieves 72.5% accuracy, with 70% sensitivity and 75% specificity. The two methods having relatively poor results also prove the superiority of the model using curvature information, which is obtained by quasiconformal theory.

### 4.3. Cross Validation on Models

Due to the limited availability of facial data and OSA index measurements for children, we opted to use the entire data set for both training and validation, for all the models in Section 4.2. As bias may be induced, 20-fold cross validation is performed to the binary-class model.

The binary model, which included 20 normal subjects and 20 subjects with severe OSA, was run twenty times, with one subject from each class being excluded from training each time. The template feature vector and detection threshold trained by the remaining 38 subjects were then used for classification of these two subjects. This process was repeated for each subject exactly once. As a result, bias can be eliminated, since subjects to be tested are not included in the training set in this cross validation process.

After incorporating cross validation, the accuracy decreased to 77.5%, with 75% sensitivity and 80% specificity. Despite the reduction in accuracy, the results are still acceptable given the extremely small data set. Furthermore, this results still demonstrated the strong discriminative power of curvatures of facial points, validating the underlying principle of surface geometry. One point to note is that since the data set is small, misclassification of a single subject could significantly impact the accuracy, sensitivity, and specificity of the model.

### 4.4. Comparison with Other Models

To emphasis the contribution of the use of 3D surfaces, a model using 2D images is built based on the same database. Similar landmarks on the jaw and chin are picked in the same sense. With the help of the MATLAB classification learner, the highest accuracy is about 50% which is far below the accuracy of our proposed model. This may follow that the local distortion information has a stronger relationship to traditional cephalometric proportions.

While some previous models already achieve similar two-class OSA classification, their subjects are most adults and 3D images obtained by X-ray or 3D scanner are required. A previous OSA classification model also used quasiconformal geometry for OSA classification [11]. However, this model has a non-negligible disadvantage. X-ray images must be taken, and landmarks are required to be manually selected by doctors, which is also time-consuming and inefficient. Our proposed model only requires a few pictures which can be taken by a mobile phone, and landmarks can be automatically selected. Moreover, our proposed model can achieve equivalent accuracy as the previous QC model. Therefore, it helps reduce the existing hospital's workload in Hong Kong.

## 5. Conclusions

This study presents a novel quasiconformal based geometric model for craniofacial analysis. The model is based on quasiconformal Teichmüller mapping and local curvature information around feature landmarks and can be used for medical classification. The model starts from few 2D images of the human face, which are used to reconstruct the 3D craniofacial structure for further analysis. The significance of each 2D image for the 3D reconstruction can be adjusted through weighting functions. The geometric information is extracted near feature landmarks, which helps to avoid unnecessary influence from meaningless regions of the structure. The LTCD is used to build the classification

machine and a non-linear balancing term is added to increase the gap between classes. More specifically, the composite score model and the parameter optimization are proposed in this work to further increase the classification accuracy. The model has been applied to study the craniofacial structures of children with and without OSA and has achieved a high accuracy of 90%, while some previous models already adopt two-class OSA classification based on adult subjects and 3D images [6,7,9,21,22], our model only requires 2D images taken by a mobile phone, which provides a preliminary selection of potential OSA patients in an inexpensive, convenient and accurate way. Furthermore, our method addresses the three-class childhood OSA classification by our composite score model. According to the first person experience of paediatrician, the shape of the lower half of face of a child can significantly reflect if children have OSA or not. This is reasonable since the upper respiratory tract is closer to the lower half than the upper half of the face. Utilizing this piece of information, we have selected landmarks from the lower half of the face to perform training and classification, which results in satisfactory classification accuracy. This demonstrates the benefits of using a landmark-based geometric approach for craniofacial structure classification.

Potential future work includes applying the model to analyze craniofacial structures to study other syndromes associated with craniofacial anomalies, as well as developing more advanced geometric models to capture geometric differences more accurately to increase the classification accuracy.

**Author Contributions:** Conceptualization, A.M.L. and L.-M.L.; methodology, M.-H.W., M.L., K.-M.T., A.M.L. and L.-M.L.; software, M.-H.W., M.L. and K.-M.T.; validation, M.-H.W., M.L. and K.-M.T.; formal analysis, M.-H.W., M.L., K.-M.T. and L.-M.L.; investigation, M.-H.W., M.L., K.-M.T., A.M.L. and L.-M.L.; resources, C.-T.A., K.C.-C.C. and A.M.L.; data curation, H.-M.Y., C.-T.A., K.C.-C.C. and A.M.L.; writing—original draft preparation, M.-H.W., M.L., K.-M.T. and L.-M.L.; writing—review and editing, M.-H.W., M.L., K.-M.T., A.M.L. and L.-M.L.; visualization, M.-H.W., M.L. and K.-M.T.; supervision, A.M.L. and L.-M.L.; project administration, A.M.L. and L.-M.L.; funding acquisition, L.-M.L. All authors have read and agreed to the published version of the manuscript.

**Funding:** L.-M.L. is supported by HKRGC GRF (Project ID: 14306721; Reference: 2130772).

**Data Availability Statement:** Data is unavailable due to privacy. Ethics approval was obtained from the Joint Chinese University of Hong Kong–New Territories East Cluster Clinical Research Ethics Committee. Written informed consents were obtained from parents or legal guidance of the participants.

**Conflicts of Interest:** The authors declare no conflict of interest.

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
