# Peer review of "A Quasiconformal-Based Geometric Model for Craniofacial Analysis and Its Application"

_axioms, doi:10.3390/axioms12040393_

Round 1

Reviewer 1 Report

In this work craniofacial morphometric analysis is performed through geometric models. The method reconstructs a 3D image from a set of 2D images, computes a feature vector and performs classification on a set of craniofacial images of children with and without the obstructive sleep apnea.

The manuscript is well written, the state-of-art analysis provides sufficient background and include relevant references, the method description should be improved, as some details are needed for the sake of reproducibility of the method, and others should be better clarified.

How many 2D images are used for reconstructing the 3D image? Which angles?

row 219: explain the criteria for image subset selection

row 226: define the criteria of weight assignment on the basis of the quality ox f the image

page 8: the description of the criteria for obtaining the feature vector is confusing.  Better clarify it.

Page 12: it is not clear if 58 or 60 subjects are included in the study

Results section is confusing and little bit hard to follow. I suggest the authors to find a tabular way for comparing the results obtained in different conditions. Why did the authors choose 567 more discriminating features? If they have tested the recognition performance for each set of features, they can plot the obtained results for each set of features and discuss it.  

Page 13: Which features are more discriminating? The authors could provide some comments on it

Recognition is performed by training the model with a set of data and testing the data on a different set of data. Do the authors have tried to split the dataset into training and test set? Which results did they obtain? This point is important, as in the conditions adopted by the authors the model can perform well on the same data, but performance can be worst on another one.

In the conclusions the authors state that:” It is worthy mentioning that some landmarks on the lower-half face were found more powerful in childhood OSA classification in statistical meaning”.  It seems that this point is not discussed in the manuscript.

Minors:

correct the typos in the abstract (cranofacial, acheives)

row 33: define the acronym AHI

row 51: provide a reference for the VVG-face dataset

row 79-80: the acronym OSA is already defined in row 64

Double check the manuscript as it contains some typos (row 86: missing ’s’).

row 222: missing reference

formula 10: missing subscripts

row 312 and 334: the LTCD acronym is already defined before

Row 404: define h

Table 1: The table is not referred in the text. Specify whether the results are related to the model that include mild OSA subjects or not.

Reviewer 2 Report

In the paper, the authors propose a geometric model for the craniofacial morphometric analysis in which they use local curvature and Teichmuller mapping. The method obtains a high accuracy. Advantages of the proposed method are: (1) it reconstructs the face surface from a set of 2D images that can be taken with any camera, e.g., a camera in mobile phone, and not from X-ray images as in other methods, (2) the landmarks are found automatically and not by a doctor as in other methods. The paper presents interesting results, but there are some minor issues that I want to raise:
1. There are some minor English language mistakes that can be eliminated by a careful reading of the text, e.g., line 101 "value" -> "valued", caption of Fig. 1 "Demonstration the 3D" -> "Demonstration of the 3D", line 284 "indexes" -> "indices", line 292 "index" -> "indices", line 330 "class" -> "classes" etc.
2. Some abbreviations are not expanded at their first occurrence in the text, e.g., LPW, AHI, 3DMM etc. On the other hand, the OSA abbreviation has been expanded in several places. It is sufficient that it is expanded at the first occurrence.
3. At the end of the introduction, a paragraph describing the content of successive sections is missing.
4. There is no citation from where Theorem 1 and 2 have been taken.
5. In Theorem 3, the authors write {l^k})_k=1^n. The closing bracket ")" is unnecessary, and should be deleted.
6. In Theorem 3, the authors use a notion of a landmark, but do not define it.
7. Line 222, a citation is missing because the authors write "(...) we adopt the method in []".
8. In some places in the text, a variable/constant is typeset using normal font and not the mathematical one (using $n$ in LaTeX), e.g., line 253 constant k, line 404 constant h.
9. In some places, a space is missing before the opening bracket.
10. In eq. (13), the notation for A and L is not explained.
11. Line 288, the reference to Fig. 3 is incorrect because it should be a reference to Fig. 2. Moreover, a closing bracket ")" is missing.
12. In some places in the text, functions such as minimum and median are not typeset on a normal level. They are typeset in subscript.
13. Figs. 3 and 4 have no reference in the text. Each figure should have at least one reference in the text. If there is no reference, then this means that this figure is unnecessary and should be deleted.
14. On pages 11-12, the authors include some images. Each image should have a number and caption, and then the reference in the text should be made according to the number. Thus, the authors should add the missing numbers, captions and appropriate references.
15. We cannot start a subsection right after starting a section, there should be at least one paragraph of text. Thus, the authors should add such a paragraph between 4.1 and 4.1.1.
16. Table 1 has no reference in the text. Each table should have at least one reference in the text.
17. In Sec. 4.3, the authors write: "our proposed model can achieve equivalent accuracy as the previous QC model". This is just an opinion of the authors because, in the paper, we do not find any results that show this. Thus, the authors should include the missing results to justify this claim.

Round 2

Reviewer 1 Report

The authors have addressed the issues I raised.